# COLA: Cross-city Mobility Transformer for Human Trajectory Simulation

## ABSTRACT

Human trajectory data produced by daily mobile devices has proven its usefulness in various substantial fields such as urban planning and epidemic prevention. In terms of the individual privacy concern, human trajectory simulation has attracted increasing attention from researchers, targeting at offering numerous realistic mobility data for downstream tasks. Nevertheless, the prevalent issue of data scarcity undoubtedly degrades the reliability of existing deep learning models. In this paper, we are motivated to explore the intriguing problem of mobility transfer across cities, grasping the universal patterns of human trajectories to augment the powerful Transformer with external mobility data. There are two crucial challenges arising in the knowledge transfer across cities: 1) *how to transfer the Transformer to adapt for domain heterogeneity*; 2) *how to calibrate the Transformer to adapt for subtly different long-tail frequency distributions of locations*. To address these challenges, we have tailored a **C**ross-city m**O**bi**L**ity tr**A**nsformer (COLA) with a dedicated model-agnostic transfer framework by effectively transferring cross-city knowledge for human trajectory simulation. Firstly, COLA divides the Transformer into the private modules for city-specific characteristics and the shared modules for city-universal mobility patterns. Secondly, COLA leverages a lightweight yet effective post-hoc adjustment strategy for trajectory simulation, without disturbing the complex bi-level optimization of model-agnostic knowledge transfer. Extensive experiments of COLA compared to *state-of-the-art* single-city baselines and our implemented cross-city baselines have demonstrated its superiority and effectiveness. Our code will be made publicly available[1].

## CCS CONCEPTS

• **Computing methodologies** → **Modeling methodologies**; **Modeling and simulation**; **Model development and analysis**;

## KEYWORDS

Human mobility, Transfer learning, Simulation, Transformer

**ACM Reference Format:**

Anonymous Author(s). 2024. COLA: Cross-city Mobility Transformer for Human Trajectory Simulation. In *Proceedings of the ACM Web Conference 2024 (WWW '24)*. ACM, New York, NY, USA, 11 pages. https://doi.org/XXXXXXX.XXXXXXX

[1]An anonymous repository is available at https://anonymous.4open.science/r/70bb3c.

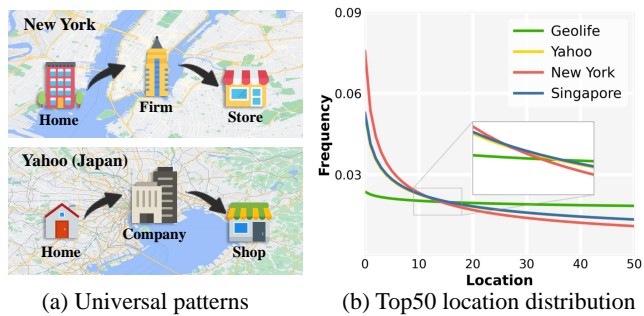

(a) Universal patterns      (b) Top50 location distribution

**Figure 1: Motivation and challenges of human trajectory simulation across cities.**

## 1 INTRODUCTION

The mobile internet has offered significant convenience for mobile devices to record individual mobility trajectories, which has shown its usage in various fields such as urban planning [30, 42], traffic control [27, 31] and epidemic prevention [29]. For example, leveraging the epidemiological model with mobility networks can conduct detailed analysis and counterfactual experiments to inform effective and equitable policy response for COVID-19 [3]. The convenience and usefulness of mobility data have also aroused the comprehensive concern of the individual privacy, which facilitates the widespread demands of human trajectory simulation [16]. Therefore, simulating human mobility behaviors necessitates modeling user intentions from a collective perspective rather than an individual one, striking a balance between generating realistic human trajectories and preserving individual privacy in the meanwhile.

Recent deep learning models [6, 7, 11, 17, 39–41] have largely promoted the synthetic quality of human trajectories based on the advanced sequence generation techniques. On the one hand, recurrent models [6, 11, 17] involve the inductive bias of human trajectory sequence. DeepMove [6] devises an attention mechanism for the long history of individual trajectories to retrieve related information; CGE [11] exploits the spatio-temporal contextual information of individual trajectories with a unified location graph. Nonetheless, recurrent models are difficult to generate the high-fidelity trajectories from scratch because they rely on historical trajectory. On the other hand, adversarial-based methods [7, 39–41] incorporate high-order semantics of human mobility such as geographical relations [7], activity dynamics [40] and Maslow's Hierarchy of Needs [41], and meanwhile maximize the long-term generation reward based on a two-player min-max game. Despite their efforts, the severe scarcity of human trajectory data will lead these dedicated models to a sub-optimal solution.

The prevalent issue of data scarcity motivates us to transfer the universal patterns of human mobility from abundant external cities to help improve the synthetic quality on our target city. As

illustrated in Fig. 1(a), daily activities of urban citizens are usually driven by similar intentions including working, entertainment, commuting, shopping, rest, etc. These common intentions exhibit universal patterns of human trajectories across different cities and result in the similar long-tail frequency distributions of locations, as shown in Fig. 1(b). The data scarcity of human trajectory can be largely alleviated if the mobility knowledge across cities can be appropriately transferred.

However, cross-city mobility transfer poses rather particular challenges compared to spatio-temporal transfer across cities [19, 25, 33], which work on air quality metrics [33], pandemic cases [25] or traffic speed [19]. Firstly, locations of the external city hardly interact with locations of the target city, causing the location embeddings non-transferrable across cities, which is called domain heterogeneity in knowledge transfer. In contrast, spatio-temporal transfer usually deals with metrics of the same feature space like air quality metrics, alleviating the transfer difficulty. Secondly, different cities present subtly different long-tail frequency distributions of locations due to the urban culture or geographical effects. The subtle differences necessitate carefully calibrating existing overconfident deep neural networks [15] during the knowledge transfer process. The aforementioned challenges require us to rethink the principles of mobility transfer across cities.

To address these challenges, we introduce the powerful Transformer [26, 28] block in a transfer learning framework to learn the universal patterns of human mobility based on the attention similarities between tokens (locations), which has demonstrated its generalization ability in many NLP tasks. Concretely, we have tailored a **C**ross-city m**O**bi**L**ity tr**A**nsformer with a model-agnostic transfer framework [8, 25], dubbed COLA, to deal with the domain heterogeneity and different long-tail frequency distributions of locations across cities. Firstly, COLA divides the Transformer into private modules accounting for city-specific characteristics and shared modules accounting for city-universal knowledge, named Half-open Transformer. It places the attention computation mechanism into shared modules to better facilitate the pattern transfer among urban human trajectories. Once transferred, the target city can exhibit its specific mobility behaviors with the private modules including non-transferrable location embeddings and their latent representations. Secondly, COLA aligns its prediction probabilities of locations with the real long-tail frequency distribution in a post-hoc manner [23] to remedy the overconfident problems [15]. Compared to the iterative optimization of re-weighted loss functions [44, 46], the post-hoc adjustment of the prediction probabilities works only for the target city after the mobility transfer is finished, leaving the minimum changes to the complex optimization of the transfer framework. COLA can effectively adapt the powerful Transformer for cross-city mobility transfer with these dedicated designs.

In conclusion, our main contributions are summarized as follows:

- We investigate the intriguing problem of cross-city human trajectory simulation, with identification of particular challenges compared to spatio-temporal transfer across cities.
- We have designed the dedicated method COLA with a model-agnostic transfer framework, which leverages our proposed Half-open Transformer to split private and shared

modules and calibrates the prediction probabilities for city-specific characteristics.
- We conduct extensive experiments on human trajectory datasets of four cities and demonstrate the superiority of COLA compared to *state-of-the-art* single-city baselines and our implemented cross-city baselines.

## 2 RELATED WORK

**Human Trajectory Simulation.** Researches on human trajectory simulation can be categorized as Markov-based methods, RNN-based methods and Adversarial-based methods. (1) Markov-based methods [10, 38] characterizing human trajectory with finite parameters of clear physical meaning are satisfactory in some cases but unable to capture complex patterns. (2) RNN-based methods [6, 11, 17] can be directly employed to generate trajectories, while they are trained for short-term goals (location prediction, which emphasizes recovering user-specific real data) and fail to generate high-quality long-term trajectories (trajectory simulation, which highlights replicating the characteristics of user-anonymous real data). (3) Adversarial-based methods [7, 39–41] can efficiently capture complex mobility patterns leveraging prior knowledge of human trajectories, while they struggle to achieve satisfactory performance on data-scarce cities.

**Cross-city Transfer Learning.** Knowledge transfer [12, 20] aims to tackle machine learning problems in data-scarce scenarios. In the field of urban computing [1, 2, 13], it's an ongoing research challenge to achieve cross-city knowledge transfer, reduction of data collection costs and higher learning efficiency. FLORAL [33] leverages multi-modal data to achieve the sing-city transfer; Region-Trans [32] divides the source city and the target city into different grids for spatio-temporal feature matching. MetaST [37] first designs a spatial-temporal network to realize multi-city knowledge transfer. Panagopoulos et al. [25] use graph representation learning to transfer cross-country knowledge between population mobility and COVID-19 transmission. TrafficTL [21] employs a periodicity-based transfer paradigm to achieve cross-city traffic prediction. Existing cross-city transfer learning methods leverage various techniques such as meta-learning and domain adaptation, however, they are limited by homogeneous domains where source domain and target domain share the same feature space. Our investigated problem involves heterogeneous domains, where the locations of source cities and the target city hold distinct meanings and cannot be directly transferred like them.

**Long-tail Learning.** The imbalance caused by long-tail distribution of labels can be mitigated by re-sampling methods [4, 22, 45], class-sensitive learning [5, 44, 46] and logit adjustment methods [18, 23, 34]. However, in cross-city human trajectory simulation task, it's crucial not only to avoid undesirable bias towards dominant labels but also accurately capture the city-specific long-tail characteristics across multiple cities.

## 3 METHODOLOGY

In this section, we propose a Cross-city mObiLity trAnsformer (COLA) for human trajectory simulation, whose framework is presented in Fig. 2. In this framework, we design Half-open Transformer dividing private and shared parameters to adapt for domain

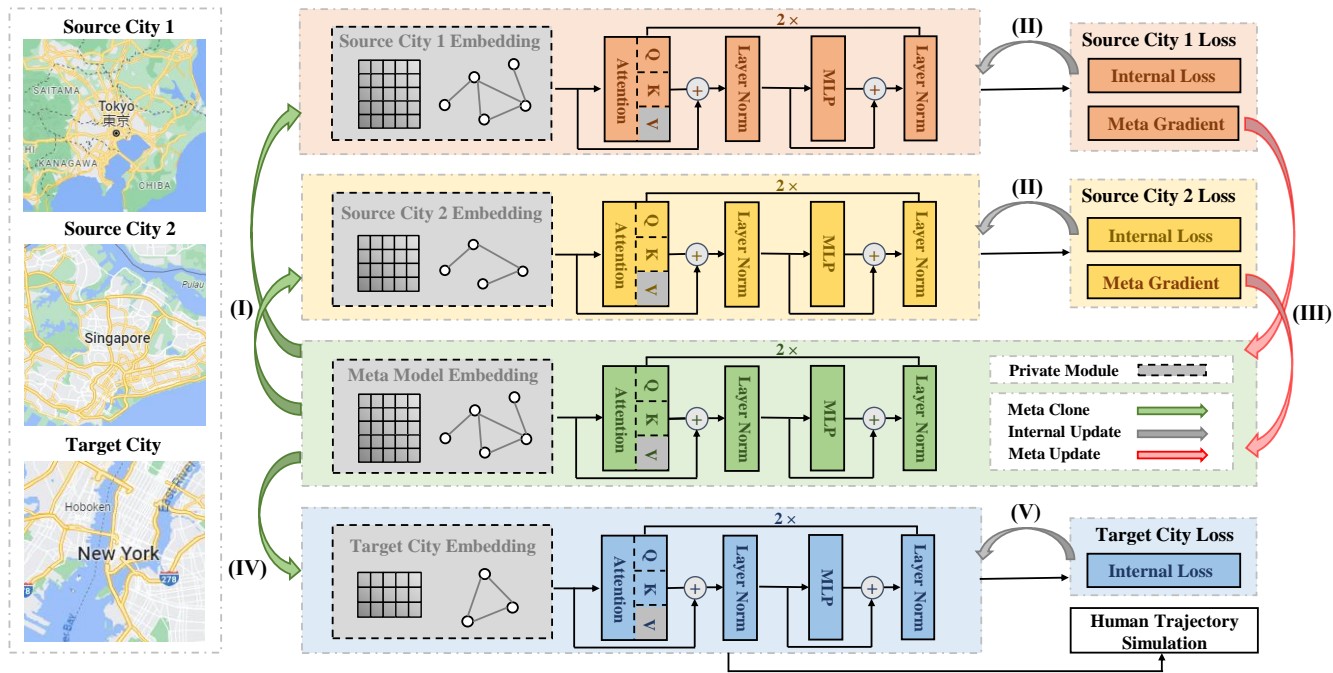

**Figure 2: The overall framework of COLA. (I) Initialize the shared parameters of the source model with the meta model. (II) Optimize the source model with its internal loss. (III) Update the meta model based on the gradient evaluated on the source city. (IV) Initialize the shared parameters of the target model with the meta model updated by all source cities. (V) Optimize the target model with its internal loss. (VI) Simulate human trajectories with Post-hoc Adjustment technique.**

heterogeneity. During simulation, we leverage Post-hoc Adjustment to calibrate the overconfident probabilities of the model, where the dynamic effects for different locations are further analyzed.

## 3.1 Overall Framework of COLA

COLA achieves the knowledge transfer across cities based on a dedicated model-agnostic transfer framework. The dataset for cross-city human trajectory simulation includes meta training set $\mathcal{M}_{\text{tr}}$ and meta test set $\mathcal{M}_{\text{te}}$. Specifically, $\mathcal{M}_{\text{tr}} = \{\mathcal{D}^k \mid k \in [K]\}$ is from $K$ source cities, where $\mathcal{D}^k = (\mathcal{D}^k_{\text{train}}, \mathcal{D}^k_{\text{test}})$, and $\mathcal{M}_{\text{te}} = \{\tilde{\mathcal{D}}\} = \{(\tilde{\mathcal{D}}_{\text{train}}, \tilde{\mathcal{D}}_{\text{test}})\}$. $\mathcal{D} = \{\mathbf{x}^m \mid m \in [M]\}$ represents real-world mobility trajectories of each city, where $M$ is the number of trajectories. Each trajectory $\mathbf{x} = (\mathbf{x}_1, \mathbf{x}_2, \cdots, \mathbf{x}_T)$ is a spatiotemporal sequence with length $T$. Our goal is leveraging common patterns learnt from trajectories of other cities to improve the quality of simulated trajectories on our target city.

Different from the classical meta learning [8] sharing all parameters of models, the parameters in our Half-open Transformer $\mathcal{T}$ are divided into private parameters from location embeddings and private attention module for city-specific characteristics, and shared parameters from shared attention module for city-universal patterns to handle domain heterogeneity. As illustrated in Fig. 2, the shared parameters of the meta model are updated by source models, which then initialize the shared parameters of the target model to effectively transfer the universal patterns from source cities to the

target city. Meanwhile, the private parameters of source models and the target model are consistently updated through *Internal Update*.

Let $\Theta_{\text{meta}}$, $\Theta_{\text{src}}$, and $\Theta_{\text{tgt}}$ be the parameters of $\mathcal{T}$ for the meta, source and target models respectively. In order to capture the universal patterns coexisting across cities, the shared parameters of each source model are initialized from the current meta model. The process of cloning the shared parameters from the meta model is named *Meta Clone*. In the framework of COLA, a source model firstly conducts *Meta Clone* as follows:

$$\Theta_{\text{src}} = \{\theta \mid \forall \theta \in \Theta^{\text{shared}}_{\text{meta}}\} \cup \{\theta \mid \forall \theta \in \Theta^{\text{private}}_{\text{src}}\}, \quad (1)$$

where $\Theta_{\text{src}}$ refers to $\Theta^k_{\text{src}}$ for simplicity. The shared parameters cloned from the meta model provide the source model with a good starting point for training. Subsequently, the source model performs *Internal Update* through the trajectory pretraining task:

$$\Theta_{\text{src}} = \Theta_{\text{src}} - \alpha_{\text{src}} \nabla_{\Theta_{\text{src}}} \mathcal{L}(\mathcal{T}_{\Theta_{\text{src}}}(\mathcal{D}^k_{\text{train}})), \quad (2)$$

where $\alpha_{\text{src}}$ is the learning rate for the source model and $\mathcal{L}$ is the loss function defined as Eq. (12). Once the optimization is complete, the source model acquires the prior knowledge of human mobility in this city. Then the meta model executes *Meta Update*, guided by the *meta gradient* evaluated on the source city, to enable the rapid adaptation of the target city:

$$\Theta_{\text{meta}} = \Theta_{\text{meta}} - \alpha_{\text{meta}} \nabla_{\Theta_{\text{src}}} \mathcal{L}(\mathcal{T}_{\Theta_{\text{src}}}(\mathcal{D}^k_{\text{test}})), \quad (3)$$

where $\alpha_{\text{meta}}$ is the learning rate for the meta model. The *Meta Clone*, *Internal Update*, and *Meta Update* are successively performed by

**Figure 3: Half-open attention in the Transformer.**

all source cities, resulting in the meta model that incorporates the universal patterns observed in all the source cities. Afterwards, the target model employs *Meta Clone* to efficiently learn the common patterns from source cities and perform *Internal Update* based on the data of the target city:

$$\Theta_{\text{tgt}} = \{\theta \mid \forall \theta \in \Theta_{\text{meta}}^{\text{shared}}\} \cup \{\theta \mid \forall \theta \in \Theta_{\text{tgt}}^{\text{private}}\}, \tag{4}$$

$$\Theta_{\text{tgt}} = \Theta_{\text{tgt}} - \alpha_{\text{tgt}} \nabla_{\Theta_{\text{tgt}}} \mathcal{L}(\mathcal{T}_{\Theta_{\text{tgt}}}(\tilde{\mathcal{D}}_{\text{train}})), \tag{5}$$

where $\alpha_{\text{tgt}}$ is the learning rate for the target model. There are three different loops $E_{\text{meta}}$, $E_{\text{src}}$ and $E_{\text{tgt}}$ for optimization. Unlike the classical meta learning where target domain updates $E_{\text{tgt}}$ epochs, $\Theta_{\text{tgt}}^{\text{private}}$ updates $E_{\text{meta}} \times E_{\text{tgt}}$ epochs to synchronize the update with meta model. After the optimization of $\Theta_{\text{tgt}}$, COLA samples $\hat{\mathbf{x}}_{t+1}$ with the following Post-hoc Adjustment:

$$\hat{\mathbf{x}}_{t+1} \sim \mathcal{T}(\mathbf{x}_{1:t})/\pi^{\tau}, \ t+1 \leq T, \tag{6}$$

where $\pi$ is the empirical location frequencies on the training samples, $\tau$ is a hyperparameter and $\tau > 0$. The details of Post-hoc Adjustment are illustrated in Section 3.3 and the overall algorithm of COLA is presented in Alg. 1.

## 3.2 Half-open Transformer

In the Half-open Transformer, in addition to location embeddings, *Value* of the attention module is set as private for domain heterogeneity adaptation, because they encapsulates the city-specific characteristics. *Query* and *Key* of the attention module compute the attention weights of a location to all locations appeared in the historical trajectory to capture city-universal patterns, suggesting their capability for sharing across cities.

Let $\mathcal{P} = \{\ell_i \mid i \in [N]\}$ denote the set of city locations $\ell$, where $N$ is the number of locations. For a trajectory $\mathbf{x} = (\mathbf{x}_1, \mathbf{x}_2, \cdots, \mathbf{x}_T)$, we discrete each location $\mathbf{x}_t$ in the trajectory and encode it with one-hot vectors: $\mathbf{h}_t = \text{Emb}^{\text{private}}(\mathbf{x}_t), t \in [T]$. Then, we map it into private and shared representations by Proj for further extracting private characteristics and shared patterns in the trajectories:

$$\begin{aligned} \mathbf{h}_t^{\text{private}} &= \text{Proj}^{\text{private}}(\mathbf{h}_t), \\ \mathbf{h}_t^{\text{shared}} &= \text{Proj}^{\text{shared}}(\mathbf{h}_t), \end{aligned} \tag{7}$$

where Proj is a projection function comprising of an identity layer or a position-wise MLP to employ the non-linear capability. Based on the distinct representation of private characteristics and shared patterns, we employ a causal self-attention mechanism to generate the trajectory sequentially. The causal attention allows locations to only attend to preceding locations observed in trajectories during training, which ensures that the model does not have access to

the future trajectory when predicting, making the training process more efficient [26]. Therefore, we project $\mathbf{h}_t^{\text{shared}}$ and $\mathbf{h}_t^{\text{private}}$ respectively with independent operations of two category into *Query* vectors $\mathbf{q}_t$, *Key* vectors $\mathbf{k}_t$ and *Value* vectors $\mathbf{v}_t$:

$$\begin{aligned} \mathbf{v}_t &= f(\mathbf{h}_t^{\text{private}}; \mathbf{W}_v), \\ (\mathbf{q}_t, \mathbf{k}_t) &= f(\mathbf{h}_t^{\text{shared}}; \mathbf{W}_q, \mathbf{W}_k). \end{aligned} \tag{8}$$

where $\mathbf{W}_q$ and $\mathbf{W}_k$ are shared parameters like $\mathbf{h}_t^{\text{shared}}$, while $\mathbf{W}_v$ are private parameters similar to $\mathbf{h}_t^{\text{private}}$, all of them are weighted matrices. After that, we utilize causal scaled dot-product attention on them to derive the attention coefficients for all appeared locations and compute the weighted sum of *Value* vectors to serve as the feature representation of the past trajectory:

$$\begin{aligned} \alpha_{tt'} &= \frac{\exp(\mathbf{q}_t \cdot \mathbf{k}_{t'}/\sqrt{d})}{\sum_{t' \leq t} \exp(\mathbf{q}_t \cdot \mathbf{k}_{t'}/\sqrt{d})}, \\ \mathbf{z}_t &= \text{MultiHead}(\sum_{t' \leq t} \alpha_{tt'} \mathbf{v}_{t'}) \mathbf{W}_o, \end{aligned} \tag{9}$$

where $d$ is the dimension of *Key* vectors and $\mathbf{W}_o$ is the weighted matrix for output. Furthermore, we use stacking operations to model the relationships from different subspaces and generate a comprehensive representation of the past trajectory. Let $\mathbf{h}_t^{L-1}$ be the representation at $L-1$-th layer. The weighed representation at this layer $\mathbf{z}_t^{L-1}$ is added to $\mathbf{h}_t^{L-1}$ and then subjected to layer normalization to obtain the intermediate representation $\bar{\mathbf{h}}_t$. Subsequently, the representation at $L$-th layer $\mathbf{h}_t^L$ can be obtained with a non-linear operation followed by another layer normalization. It can be expressed as follows:

$$\begin{aligned} \bar{\mathbf{h}}_t &= \text{LayerNorm}(\mathbf{h}_t^{L-1} + \mathbf{z}_t^{L-1}), \\ \mathbf{h}_t^L &= \text{LayerNorm}(\text{MLP}(\bar{\mathbf{h}}_t)). \end{aligned} \tag{10}$$

Then we use a linear layer to process the feature to obtain the logits of all locations. The logit at location $\ell_i$ can be expressed $\mathbf{p}_{t+1}^{m,i} = \omega_i^{\text{private}} \mathbf{h}_t^L$, where $\omega_i^{\text{private}}$ is private and same as the weights of $\ell_i$ embedding. Furthermore, the predicted probability of location $\ell_i$ can be obtained after normalizing:

$$\hat{\mathbf{y}}_{t+1}^{m,i} = \frac{\exp(\mathbf{p}_{t+1}^{m,i})}{\sum_{j \in [N]} \exp(\mathbf{p}_{t+1}^{m,j})}, \tag{11}$$

which is used for the post-hoc sampling in the simulation phase. During the training phase, the Transformer $\mathcal{T}$ is optimized with the following *internal loss* function:

$$\mathcal{L} = -\frac{1}{MT} \sum_{m=1}^{M} \sum_{t=1}^{T-1} \mathbf{y}_{t+1}^m \cdot \log(\hat{\mathbf{y}}_{t+1}^m). \tag{12}$$

## 3.3 Post-hoc Adjustment

On the one hand, as illustrated in Fig. 1 (b), the overall visited frequency distributions of locations appeared in trajectories from four cities all exhibit long-tailed characteristics. Under the circumstances, some locations are visited occasionally but cannot be disregarded. Nevertheless, the paucity of occasionally visited locations poses a significant challenge in terms of generalisation. Furthermore, naive learning on such data is susceptible to an undesirable

bias towards dominant locations [23]. Due to the prevalent overconfident problem of deep learning models [15], our proposed Transformer even exaggerates the real long-tailed frequency distribution of locations that severely overlooks the low-frequency locations.

On the other hand, the long-tailed visited frequency distributions of locations from multiple cities display subtly differences, especially in the enlarged part of Fig. 1 (b), which is crucial in determining the appropriate long-tail learning method for addressing the class imbalance problem. Traditional long-tail learning methods encompass two strands of work: post-hoc normalisation of class weights and loss modification to account for varying class penalties. Because loss modification requires loss functions with different weights for models of various cities, which hinders the transfer of universal mobility patterns across cities and also increase the training cost, it is unsuitable for calibrating the imbalance of locations in trajectories during training. Therefore, COLA leverages the post-hoc adjustment method to calibrate our overconfident probabilities of locations instead of pursuing the optimal Bayesian error of long-tailed learning [23], written as follows:

$$\tilde{y}_{t+1}^{m,i} = \frac{\exp(\mathbf{p}_{t+1}^{m,i})/\pi_i^\tau}{\sum_{j\in[N]} \exp(\mathbf{p}_{t+1}^{m,j})/\pi_j^\tau}, \tau > 0. \quad (13)$$

The penalization term $\pi_i^\tau$ towards high-frequency locations offers a lightweight yet effective way to simulate the real long-tailed frequency distribution of locations. During the simulation phase, we sample $\hat{\mathbf{x}}_{t+1}^m \sim \tilde{y}_{t+1}^m$ from $\mathcal{T}(\mathbf{x}_{1:t}^m)$ following the above calibrated probabilities of locations.

PROPOSITION 1. *Suppose that the probability density function of locations follows Zipf's law $\pi(x) \sim ax^{-\gamma}, \gamma > 0, x \in \mathbb{N}^+$ is the index of a location, the post-hoc adjustment dynamically scales the pair-wise probabilities of two locations as: $\frac{\tilde{y}_{t+1}^{m,i}}{\tilde{y}_{t+1}^{m,j}} = \frac{\hat{y}_{t+1}^{m,i}}{\hat{y}_{t+1}^{m,j}} \cdot (i/j)^{\tau\cdot\gamma}$.*

Let $i < j$ for simplicity, where $i, j$ are indices of locations sorted by their frequencies. It indicates that high-frequency locations are penalized in proportion to their frequency, while low-frequency locations are rewarded in contrast. In situations where both locations exhibit similar frequencies, the unadjusted probabilities of the models play a crucial role.

## 4 EXPERIMENT

To demonstrate the effectiveness of the proposed COLA method, we conduct extensive experiments to answer the following research questions:

- **RQ1**: How does COLA perform compared to *state-of-the-art* single-city baselines and our implemented cross-city baselines in human trajectory simulation task?
- **RQ2**: How do different components of COLA contribute to the final performance?
- **RQ3**: Can COLA generate synthetic data of high quality for practical applications?
- **RQ4**: How do hyperparameter settings influence the performance of COLA?

---

**Algorithm 1** COLA

**Require:** source cities and the target city dataset: $\mathcal{M}_{tr}$ and $\mathcal{M}_{te}$; learning rates: $\alpha_{src}, \alpha_{meta}, \alpha_{tgt}$; Training epochs: $E_{meta}, E_{src}, E_{tgt}$.

1: Initialize $\Theta_{meta}$ randomly
2: **for** epoch $e_m \in E_{meta}$ **do**
3:     **for** each city $\mathcal{D}^k \in \mathcal{M}_{tr}$ **do**
4:         Meta Clone for $\Theta_{src}^k$ with Eq. (1)
5:         # Iterate $E_{src}$ epochs to update $\Theta_{src}^k$
6:         Internal Update for $\Theta_{src}^k$ with Eq. (2)
7:         Meta Update for $\Theta_{meta}$ with Eq. (3)
8:     **end for**
9:     Meta Clone for $\Theta_{tgt}$ with Eq. (4)
10:     # Iterate $E_{tgt}$ epochs to update $\Theta_{tgt}$
11:     Internal Update for $\Theta_{tgt}$ with Eq. (5)
12: **end for**
13: Perform simulation with Eq. (6)

---

**Table 1: The statistics of four datasets.**

| Dataset | # Users | # Locations | # Visits | # AvgStep/Day |
|---------|---------|-------------|----------|---------------|
| Geolife | 153 | 32,675 | 34,834 | 8.9 |
| Yahoo | 10,000 | 16,241 | 188,061 | 18.8 |
| New York | 1,189 | 9,387 | 19,040 | 6.9 |
| Singapore | 1,461 | 11,509 | 38,522 | 7.0 |

### 4.1 Experimental Setup

*4.1.1 Datasets.* We evaluate the performance of COLA and baselines on four publicly available datasets:

- **Geolife** [47]: GeoLife consists of 17,621 trajectories collected by 182 users over a peroid of five years (from April 2007 to August 2012), which are primarily located in Beijing, China.
- **Yahoo (Japan)** [35][2]: The dataset contains 100K individuals' trajectories across 90 days in a metropolitan area provided by Yahoo Japan Corporation, which is called Yahoo for simplicity.
- **New York** and **Singapore** [36]: The two datasets are derived from Foursquare[3], which captures user behavior worldwide. We select the records from New York and Singapore within it.

The four datasets are preprocessed following the protocol [7] of human trajectory simulation. For convenience and universality of modeling, an hourly time slot is adopted as the basic simulation unit. Moreover, only the trajectories with at least six visit records per day are considered in our study. For Yahoo, we select 10,000 high-quality trajectories from its preprocessd data. The data statistics of preprocessed datasets are presented in Table 1. The daily average length of trajectories in New York and Singapore is generally smaller than that in GeoLife and Yahoo, which aggravates the irregularity and thereby increases the difficulty of modeling. For each dataset, the training, valid and test sets follow a ratio of 7:1:2.

*4.1.2 Baselines.* We compare the proposed COLA with the following two categories of *state-of-the-art* baselines:

---

[2]https://connection.mit.edu/humob-challenge-2023
[3]https://foursquare.com/

**Table 2: The performance comparison between COLA and baselines for human trajectory simulation. All experimental results are conducted over five trials for a fair comparison. Note that a lower JSD value indicates a better performance. AVG is the average rank across six metrics. Bold and underline mean the best and the second-best results. "*" implies statistical significance for $p < 0.05$ under paired t-test.**

| Dataset | Geolife | | | | | | | Yahoo | | | | | | |
|---|---|---|---|---|---|---|---|---|---|---|---|---|---|---|
| Metrics (JSD)↓ | Distance | Radius | Duration | DailyLoc | G-rank | I-rank | AVG↓ | Distance | Radius | Duration | DailyLoc | G-rank | I-rank | AVG↓ |
| Markov | 0.0316 | 0.1341 | 0.0135 | 0.1424 | 0.0830 | 0.0931 | 8.7 | 0.2008 | 0.5037 | 0.1682 | 0.5031 | 0.0406 | 0.0574 | 7.7 |
| IO-HMM | 0.2384 | 0.4138 | 0.0100 | 0.1124 | 0.1110 | 0.0799 | 9.2 | 0.5690 | 0.6772 | 0.0847 | 0.6716 | 0.0388 | 0.1031 | 10.5 |
| DeepMove | 0.2064 | 0.3743 | 0.0099 | 0.1269 | 0.1805 | 0.0584 | 7.3 | 0.5400 | 0.6775 | 0.0622 | 0.6061 | 0.2590 | 0.1079 | 10.5 |
| GAN | 0.2224 | 0.3887 | 0.0108 | 0.1114 | 0.0812 | 0.0590 | 7.0 | 0.5705 | 0.6786 | 0.0852 | 0.6748 | 0.0400 | 0.0989 | 11.2 |
| MoveSim | 0.2203 | 0.3851 | 0.1292 | 0.1424 | 0.0860 | 0.0736 | 9.8 | 0.4579 | 0.5065 | 0.0767 | 0.5843 | 0.0331 | 0.0838 | 7.5 |
| CGE | 0.3636 | 0.6637 | 0.0107 | 0.1125 | 0.0783 | 0.0768 | 8.5 | 0.3060 | 0.5610 | 0.0853 | 0.5339 | 0.0474 | 0.0740 | 9.0 |
| ACT-STD | 0.2007 | 0.3613 | 0.6931 | 0.6931 | 0.1752 | 0.0629 | 10.0 | 0.1438 | 0.3965 | 0.0839 | 0.1280 | 0.6931 | 0.0740 | 7.2 |
| LSTM | 0.0305 | 0.1224 | 0.0133 | 0.1496 | 0.0799 | 0.0595 | 6.8 | 0.0352 | 0.1702 | 0.0011 | 0.0231 | 0.0326 | 0.0633 | 3.3 |
| SeqGAN | 0.0383 | 0.1294 | 0.0100 | 0.1140 | 0.0793 | 0.0562 | 5.0 | 0.2906 | 0.2614 | 0.0255 | 0.1576 | 0.0335 | 0.0608 | 4.8 |
| MobFormer | 0.0300 | 0.1220 | 0.0278 | 0.2481 | **0.0776** | 0.0911 | 6.8 | 0.2323 | 0.3646 | 0.1423 | 0.4497 | 0.0348 | 0.0552 | 6.2 |
| CrossLSTM | 0.0300 | 0.1219 | 0.0097 | 0.1118 | 0.0788 | 0.0566 | 2.8 | 0.0223 | 0.1695 | 0.0004 | 0.0033 | 0.0358 | 0.0458 | 2.7 |
| CrossSeqGAN | 0.2315 | 0.4083 | 0.0110 | 0.1118 | 0.0794 | 0.0597 | 7.8 | 0.5166 | 0.6656 | 0.0718 | 0.6383 | 0.0400 | 0.1031 | 9.5 |
| COLA | **0.0294*** | **0.1205*** | **0.0096** | **0.1103*** | 0.0780 | **0.0559*** | **1.2** | **0.0161*** | **0.1462*** | **0.0002*** | **0.0029*** | **0.0294*** | **0.0447*** | **1.0** |

| Dataset | New York | | | | | | | Singapore | | | | | | |
|---|---|---|---|---|---|---|---|---|---|---|---|---|---|---|
| Metrics (JSD)↓ | Distance | Radius | Duration | DailyLoc | G-rank | I-rank | AVG↓ | Distance | Radius | Duration | DailyLoc | G-rank | I-rank | AVG↓ |
| Markov | 0.0354 | 0.0917 | 0.0060 | 0.0595 | 0.1470 | 0.0693 | 8.0 | 0.0245 | 0.0462 | 0.0097 | 0.1506 | 0.1793 | 0.0631 | 9.0 |
| IO-HMM | 0.1630 | 0.3283 | 0.0055 | 0.1166 | 0.1615 | 0.0624 | 10.8 | 0.1839 | 0.2524 | 0.0066 | 0.1597 | 0.2000 | 0.0612 | 11.0 |
| DeepMove | 0.1632 | 0.3216 | 0.0006 | 0.0266 | 0.1501 | 0.0554 | 8.5 | 0.1850 | 0.2569 | 0.0011 | 0.0507 | 0.1031 | 0.0541 | 7.5 |
| GAN | 0.1601 | 0.3010 | 0.0046 | 0.0963 | 0.0914 | 0.0584 | 9.0 | 0.1929 | 0.2543 | 0.0059 | 0.1316 | 0.1688 | 0.0556 | 10.3 |
| MoveSim | 0.1320 | 0.2365 | 0.0067 | 0.3241 | 0.0813 | 0.0461 | 8.2 | 0.0950 | 0.1411 | 0.0043 | 0.3731 | 0.1220 | 0.0539 | 7.3 |
| CGE | 0.1486 | 0.4002 | 0.0243 | 0.4170 | 0.0995 | 0.0438 | 9.8 | 0.1341 | 0.4535 | 0.1279 | 0.5706 | 0.1636 | **0.0524** | 9.5 |
| ACT-STD | 0.1428 | 0.3341 | 0.4554 | 0.6346 | 0.1420 | 0.0479 | 10.5 | 0.0895 | 0.1758 | 0.3271 | 0.5236 | 0.1820 | 0.0542 | 9.7 |
| LSTM | 0.0265 | 0.0922 | 0.0015 | 0.0322 | 0.0084 | 0.0423 | 3.7 | 0.0072 | 0.0271 | 0.0008 | 0.0181 | 0.0146 | 0.0555 | 4.0 |
| SeqGAN | 0.0338 | 0.1088 | 0.0068 | 0.1396 | 0.0154 | 0.0527 | 7.7 | 0.0670 | 0.1046 | 0.0020 | 0.0384 | 0.0146 | 0.0541 | 4.8 |
| MobFormer | 0.0264 | 0.0917 | 0.0005 | 0.0112 | 0.0080 | 0.0508 | 3.2 | 0.0069 | 0.0269 | 0.0053 | 0.0593 | 0.0243 | 0.0555 | 5.7 |
| CrossLSTM | 0.0266 | 0.0916 | 0.0016 | 0.0271 | 0.0091 | 0.0457 | 4.0 | 0.0070 | 0.0269 | 0.0009 | 0.0095 | 0.0181 | 0.0551 | 3.7 |
| CrossSeqGAN | 0.1257 | 0.2332 | 0.0034 | 0.0776 | 0.0089 | 0.0530 | 6.7 | 0.1701 | 0.2262 | 0.0040 | 0.1040 | **0.0069** | 0.0555 | 7.2 |
| COLA | **0.0263*** | **0.0911*** | **0.0004** | **0.0111*** | **0.0078*** | **0.0368*** | **1.0** | **0.0067*** | **0.0267*** | **0.0007*** | **0.0086*** | 0.0141 | 0.0534 | **1.3** |

- *Single-City Baselines*: **Markov** [10] is a well-known probability method, which treats locations of trajectories as states and calculates transition probabilities of locations. **IO-HMM** [38] is an extension of the traditional Hidden Markov Model (HMM). **LSTM** [17] is a basic recurrent model for sequence prediction. **DeepMove** [6] is a recurrent network with history attention. **GAN** [14] is a prominent generative framework, where both generator and discriminator are performed by two LSTMs in our setting. **SeqGAN** [39] extends GAN by using a stochastic policy in reinforcement learning (RL) to address the challenge of sequence generation. **MoveSim** [7] is an extension to SeqGAN by introducing the prior knowledge of urban structure and designing two loss functions. **CGE** [11] is a static graph-based representation for human motion built by check-ins reflecting users' geographical preferences and visiting intentions. **ACT-STD** [40] captures the spatiotemporal dynamics underlying trajectories with neural differential equations for human activity modelling. **MobFormer** is directly implemented by a Transformer [28] to highlight its capability in contrast to existing recurrent models.
- *Cross-City Baselines*: To conduct a comprehensive comparison using cross-city dataset, we further incorporate the two most effective baselines, LSTM and SeqGAN, into the proposed transfer learning framework, called **CrossLSTM** and **CrossSeqGAN**.

*4.1.3 Evaluation Metrics.* We adopt six standard metrics as employed in previous works [7, 24] to assess the quality of simulated outcomes. These metrics calculate the mobility trajectory distributions from various perspectives: (1) *Distance*: moving distance of each adjacent locations in individual trajectories (spatial perspective); (2) *Radius*: root mean square distance from a location to the center of its trajectory (spatial perspective); (3) *Duration*: dwell duration among locations (temporal perspective); (4) *DailyLoc*: proportion of unique daily visited locations to the length of the trajectory for everyone (preference perspective); (5) *G-rank*: visited frequency of the top-100 locations (preference perspective); (6) *I-rank*: the individual version of G-rank (preference perspective). We further use Jensen-Shannon divergence (JSD) [9] to measure the discrepancy of the distributions between simulated and real-world trajectories, which is defined as:

$$\text{JSD}(p\|q) = H((p+q)/2) - \frac{1}{2}(H(p) + H(q)), \quad (14)$$

where $p$ and $q$ are two compared distributions, and $H$ is the entropy. A lower JSD indicates a closer match to the statistical characteristics, suggesting a superior simulated outcome. Specifically, we calculate the average rank, denoted as *AVG*, across these six metrics for a clear comparison.

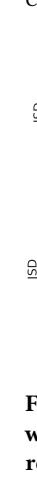

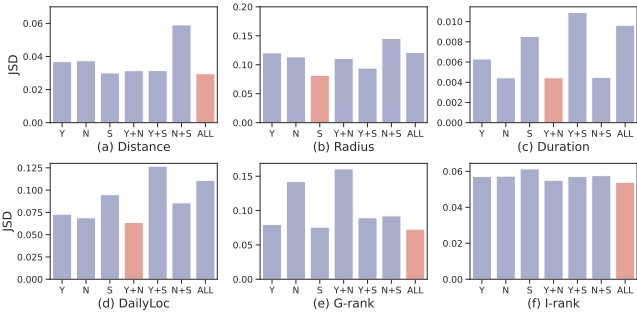

**Figure 4: The performance comparison of COLA on Geolife with different combinations of source cities. All experimental results are conducted over five trials for a fair comparison.**

*4.1.4 Experiment Settings.* (1) The single-city baselines are trained with 250 epochs. (2) The cross-city baselines are trained with 5, 1 and 50 epochs for meta, source city and target city updating, where the training epochs of the target city is in line with single-city baselines. When a city serves as the target city, the remaining three cities are considered as the source cities. The learning rate for the city models (both source and target cities) and the meta model is set to 1e-3 and 5e-4, respectively. (3) All baselines use a hidden dimension of 96 and a batch size of 32. The other specific hyperparameters of the baselines follow the settings reported in their respective papers. For COLA, the number of linear layers in MLP for extracting private and shared patterns is searched over {1, 2, 3}, and the coefficient $\tau$ for post-hoc adjustment is searched over {0.001, 0.01, 0.1, 0.25, 0.5, 1}.

## 4.2 RQ1: Overall Performance

In this section, we compare our model with the sing-city baselines and our implemented cross-city baselines over the four real-world city datasets in the human trajectory simulation task. We present the average performance of each method under five trials in Table 2. The results yield the following observations:

- *COLA steadily outperforms baselines.* COLA achieves the best AVG on all cities with ranking 1st on 21 metrics and ranking 2nd on 3 metrics over twenty-four metrics of the four datasets. It's worth noting that COLA consistently ranks first on New York with the lowest number of trajectories and Yahoo with the most, which validates the simulated data generated by our framework exhibits exceptional fidelity from spatial, temporal and preference perspective.

- *Cross-city baselines exhibit distinct results.* CrossLSTM significantly improves the performance of LSTM on almost all datasets, especially on Geolife where AVG increases from 6.8 to 2.8. However, CrossSeqGAN deteriorates the performance of SeqGAN especially on Yahoo where AVG drops from 4.8 to 9.5, which is due to its difficulty of convergence with meta-learning, unlike LSTM without generative adversarial framework.

- *Transformer is a highly promising model for human trajectory simulation.* MobFormer, a causal-attention Transformer, averagely surpasses 94% single-city baselines on four cities. On New York, it even surpasses two cross-city baselines with a reduction of

**Table 3: Ablation study on half-open attention (HA) and post-hoc adjustment (PO). Bold means the best result.**

| Dataset | Method | Distance | Radius | Duration | DailyLoc | G-rank | I-rank |
|---|---|---|---|---|---|---|---|
| Geolife | w/o HA | 0.0347 | 0.1443 | 0.0126 | 0.1498 | 0.0914 | 0.0648 |
| | w/o PO | 0.0344 | 0.1427 | 0.0138 | 0.1329 | 0.0872 | 0.0634 |
| | COLA | **0.0296** | **0.1213** | **0.0096** | **0.1110** | **0.0772** | **0.0559** |
| Yahoo | w/o HA | 0.0260 | 0.1553 | 0.0004 | 0.0043 | 0.0330 | 0.0504 |
| | w/o PO | 0.0258 | 0.1526 | 0.0003 | 0.0039 | 0.0360 | 0.0502 |
| | COLA | **0.0161** | **0.1462** | **0.0002** | **0.0029** | **0.0294** | **0.0447** |
| New York | w/o HA | 0.0275 | 0.0957 | 0.0004 | 0.0137 | 0.0112 | 0.0416 |
| | w/o PO | 0.0270 | 0.0949 | 0.0005 | 0.0124 | 0.0108 | 0.0451 |
| | COLA | **0.0263** | **0.0911** | **0.0004** | **0.0111** | **0.0078** | **0.0368** |
| Singapore | w/o HA | 0.0130 | 0.0394 | **0.0006** | 0.0103 | 0.0235 | 0.0555 |
| | w/o PO | 0.0101 | 0.0357 | 0.0027 | 0.0353 | 0.0196 | 0.0551 |
| | COLA | **0.0067** | **0.0267** | 0.0007 | **0.0086** | **0.0141** | **0.0534** |

JSD up to 69%. It indicates that the attention mechanism which captures the global attention to the history sequences are more suitable for the human trajectory simulation task.

- *The limitation of single-city baselines.* Markov performs well in spatial metrics but poorly in temporal and preference metrics. Most baselines underperform due to their excessive annotation requirements which is unsuitable for data-scarce scenario, e.g. IO-HMM and ACT-STD, or due to the model's overcomplexity which hinders its convergence, e.g. GAN, MoveSim and CGE. Nevertheless, LSTM capturing long-term dependencies and Seq-GAN based on policy gradient achieve superior performance.

## 4.3 RQ2: Ablation Studies

*4.3.1 Performance across different cities.* We conduct experiments on various source cities to explore the performance of transfer learning across cities. Fig. 4 shows the results of selecting Geolife as the target city (see more details of the other cities in A.1). For simplicity, Geolife, Yahoo, New York and Singapore are denoted as 'G', 'Y', 'N' and 'S', respectively. Obviously, leveraging all cities leads to a steady and better performance across almost all metrics, indicating that multi-city knowledge transfer can enhance the model generalizability. Specifically, the combination of Yahoo and New York achieves the best performance on *Duration* and *Dailyloc* metrics, likely due to their similarity of work habits influenced by economic development degree. different cultural traditions within cities. However, it is worth noting that the combination of New York and Singapore exhibits the worst performance on *Distance* and *Radius* metrics. This discrepancy may be attributed to their varying geographical layouts and commuting distances.

*4.3.2 Half-open Attention and Post-hoc Adjustment.* In this part, we investigate the effectiveness of the half-open attention and the post-hoc adjustment in COLA. As shown in Table 3, our model without the two modules still outperforms the best-performed baselines presented in Table 2, which demonstrates the power of our basic framework (i.e., Transformer with transfer learning). Specifically, removing the half-open attention module leads to a noteworthy deterioration on almost all metrics, especially on the metrics from spatial and preference perspective, which indicates that it's necessary to distinguish city-specific characteristics and city-universal mobility patterns when transferring. In addition, when the post-hoc adjustment module is removed, the performance significantly

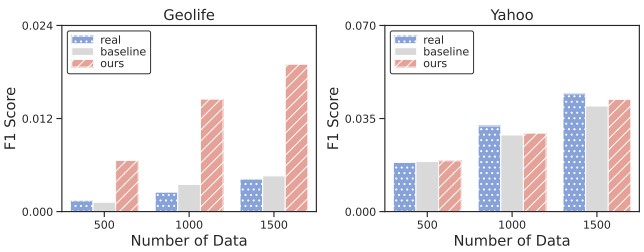

**Figure 5: Location prediction in the fully simulated scenario.**

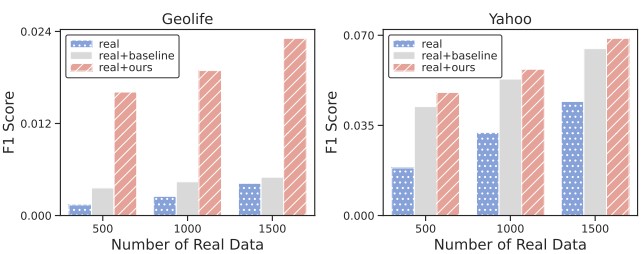

**Figure 6: Location prediction in the hybrid scenario. For scenarios with different numbers of real-world trajectories (i.e. 500, 1000, 1500), additional 1000 simulated trajectories are included for data augmentation.**

declines on almost all metrics, especially on *Duration* metric, which suggests that post-hoc adjustment effectively corrects the deviation caused by the overconfident deep models thus improving the simulation credibility for highly visited locations. Furthermore, the overall performance of the model degrades further when removing the half-open attention module.

### 4.4 RQ3: Practical Applications

In applications that depend on individual trajectories, directly sharing real location records is often infeasible due to privacy concerns. In such cases, COLA can be used to generate simulated data that can mask sensitive information while preserving the availability of real data. To evaluate the effectiveness of simulated individual trajectories, we conduct experiments with two categories of simulated data: (1) fully simulated scenarios for enhanced privacy protection, and (2) hybrid scenarios (combines real and simulated data) for data augmentation. We choose location prediction as a representative application [24, 43] which serves as the foundation for various trajectory-related problems, including location recommendation and planning. In addition, we employ a widely used LSTM model with attention mechanism to predict future locations based on historical trajectories.

As depicted in Fig. 5, for Geolife which is relatively sparser than Yahoo, simulated high-quality trajectories with less noise based on the real data enables better identification of mobility patterns, yielding superior outcomes compared to the real data. Moreover, our framework outperforms the best baseline, indicating the superiority of our method. For Yahoo, the performance of our framework aligns more closely with the real data compared to the best baseline, showcasing the utility of our simulated data. Fig. 6 illustrates that

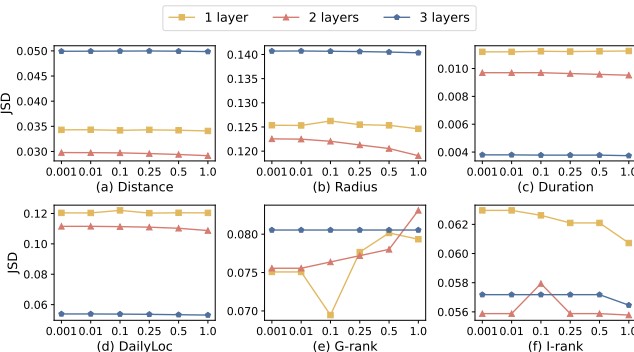

**Figure 7: The performance comparison of COLA on Geolife using different layers and post-hoc coefficients $\tau$. All experimental results are conducted over five trials.**

the model using combined data achieves better performance than only using real data. In the hybrid scenario, COLA also surpasses the best baseline. Additionally, the combination of real data and simulated data generated by COLA improves performance as the size of real data increases. The above experiments demonstrate the practical benefits of simulated human trajectories.

### 4.5 RQ4: Parameter Sensitivity

We investigate two crucial hyperparameters in COLA: the number of linear layers in the projection function of the attention module and the coefficient $\tau$ for post-hoc adjustment during trajectory simulation. Grid searches are performed over {1, 2, 3} and {0.001, 0.01, 0.1, 0.25, 0.5, 1.0} for these two hyperparameters, considering all metrics across the four datasets. Fig. 7 demonstrates the robust performance of COLA across various hyperparameter settings, indicating that different values of hyperparameters do not diminish its superiority over the baselines. Specifically, using two layers can yield satisfactory results, possibly because increasing the depth often results in overfitting, reducing the model generalizability. Conversely, decreasing the depth is prone to underfitting, which fails to fully capture the mobility patterns in trajectories. Furthermore, for Geolife, using a smaller coefficient (i.e. 0.25) for post-hoc adjustment can lead to improved performance across most metrics.

## 5 CONCLUSION

In this paper, we have tailored a Cross-city mObiLity trAnsformer with a model-agnostic transfer framework called COLA to simulate human trajectories, which tackles domain heterogeneity and overcomes the overconfident problem of deep models. Extensive experimental results demonstrate the superiority of COLA over *state-of-the-art* single-city and our implemented cross-city baselines. Nonetheless, there lacks a unified large mobility model upon millions of human trajectory data from global cities due to domain heterogeneity. Inspired by the remarkable progress of GPT, we will explore the pre-training potential of Transformer for human trajectory simulation task in the future.

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

# A EXPERIMENT RESULTS

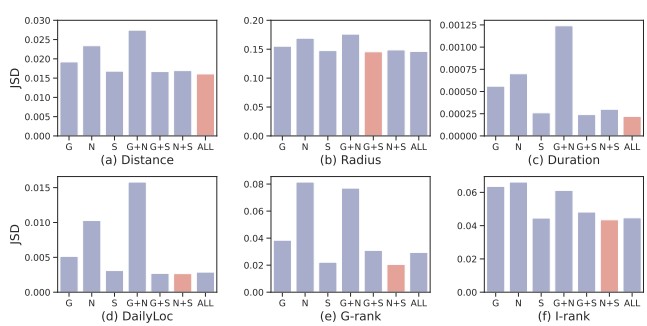

**Figure 8: The performance comparison of COLA on Yahoo with different combinations of source cities. All experimental results are conducted over five trials for a fair comparison.**

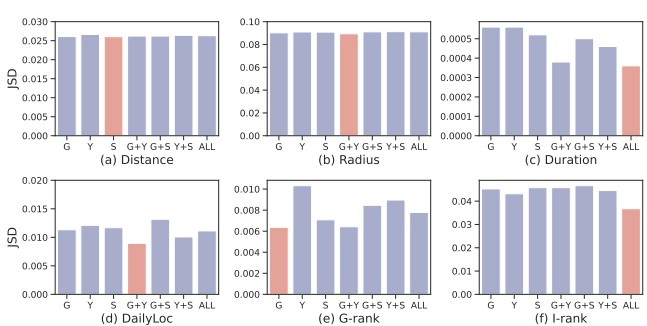

**Figure 9: The performance comparison of COLA on New York with different combinations of source cities. All experimental results are conducted over five trials for a fair comparison.**

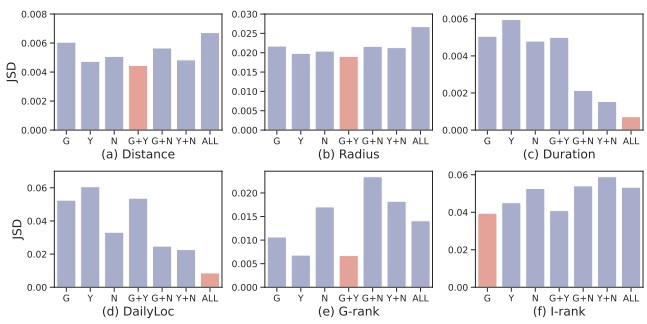

**Figure 10: The performance comparison of COLA on Singapore with different source cities. All experimental results are conducted over five trials for a fair comparison.**

## A.1 Performance Across Different Cities

we present the results of transferring with various source cities for Yahoo, New York and Singapore in Fig. 8, Fig. 9 and Fig. 10 respectively. Generally, transfer with three source cities exhibits

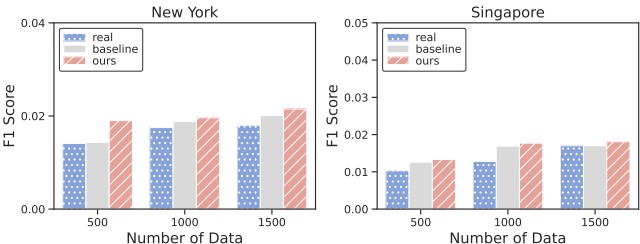

**Figure 11: Location prediction in fully simulated scenarios.**

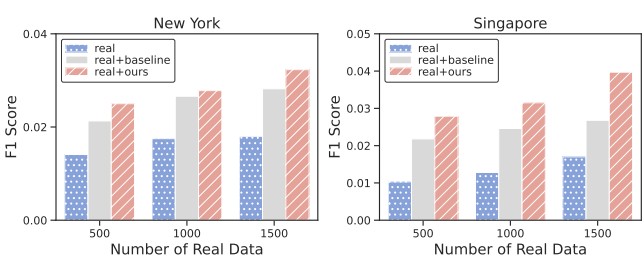

**Figure 12: Location prediction in hybrid scenarios. For scenarios with different numbers of real-world trajectories (i.e. 500, 1000, 1500), additional 1000 simulated trajectories are included for data augmentation.**

most stable and best performance on six metrics. Specifically, for Yahoo, utilizing the remaining three cites as source cities yields superior results on *Distance* and *Duration* metrics due to the complementarity of spatial and temporal patterns in their trajectories; for New York, leveraging three cities for transfer surpasses on *Duration* and *I-rank* metrics because of the effective capturing of temporal patterns and personal preferences in trajectories; for Singapore, the average performance using three source cities on six metrics is also better than other combinations of source cities, especially on *Duration* and *DailyLoc*, suggesting the effective capturing of temporal patterns in trajectories.

It's noteworthy that while some combinations of source cities achieve the optimal results on one or several metrics, their performance is less consistent and worse on more metrics. For example, transferring from Singapore to New York outperforms on *Distance* metric but demonstrates the poorest average performance on the other five metrics compared to other combinations of source cities. Nevertheless, all results using cross-city knowledge significantly outperform the baselines of this paper, validating the effectiveness of cross-city transfer learning for human mobility simulation.

## A.2 Practical Applications

The practical application results in fully simulated and hybrid scenarios for New York and Singapore are illustrated in Fig. 11 and Fig. 12. As depicted in Fig. 11, both our framework and the best baseline surpass the real data by capturing more significant mobility patterns with less noise, emphasizing the utility of the simulated data. Fig. 12 illustrates that the model using combined data obtains better performance than only using real data. Furthermore, COLA also outperforms the best baseline in hybrid scenarios. In addition,

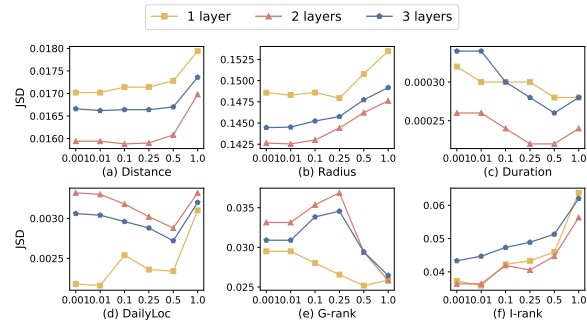

**Figure 13: The performance comparison of COLA on Yahoo using different layers and post-hoc coefficient $\tau$. All experimental results are conducted over five trials.**

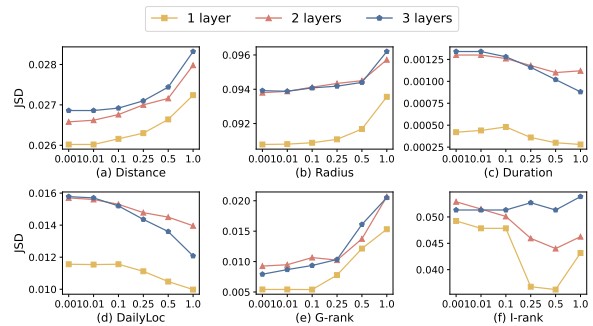

**Figure 14: The performance comparison of COLA on New York using different layers and post-hoc coefficient $\tau$. All experimental results are conducted over five trials.**

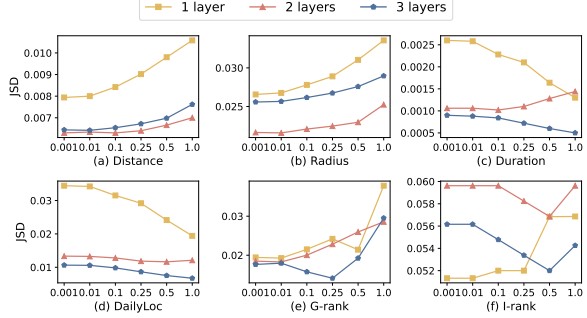

**Figure 15: The performance comparison of COLA on Singapore using different layers and post-hoc coefficient $\tau$. All experimental results are conducted over five trials.**

the combination of real data and the simulated data generated by COLA improves the performance as the size of real data increases. The above experiments showcase the practical benefits of simulated human trajectories.

## A.3 Parameter Sensitivity

The grid search results for two crucial hyperparameters (layers and post-hoc coefficients $\tau$) in COLA on Yahoo, New York and Singapore are presented in Fig. 13, Fig. 14 and Fig. 15, respectively. For Yahoo, the model with two projection layers and a coefficient of 0.5 yields superior performance. For New York, leveraging a small coefficient ($\tau = 0.25$) can exhibit good performance. Meanwhile, using one layer enables the extraction of both private and shared representation without overfitting. For Singapore, a deeper layer produces improved results for better non-linear representation. Additionally, setting $\tau = 0.25$ can better adjust the exaggerated probability caused by long-tail frequency distribution of locations.

