# OpenReview forum: "COLA: Cross-city Mobility Transformer for Human Trajectory Simulation"
_ACM.org/TheWebConf/2024/Conference — TheWebConf24_

### Official Review · Reviewer_fat5 · 2023-11-22

**Novelty:** 5
**Technical Quality:** 6

**Review:**

This research addresses the challenges in simulating human trajectories across cities. The authors propose a novel method called COLA, which overcomes the limitations of previous approaches.

The background of the study is centered around the need to simulate realistic human movement patterns across different urban environments. Existing methods have limitations in terms of capturing universal patterns and accurately representing the top location distributions. The essential difference of the proposed method lies in its ability to learn cross-city mobility transformations through neural networks, enabling the generation of diverse and realistic trajectories.

The research method proposed in this article involves training a generative model on large-scale GPS trajectories from various cities. The experiments conducted demonstrate the effectiveness of the COLA method, as it performs well in terms of trajectory similarity metrics and captures the diversity and realism of human movement across cities. In summary, this paper is well-organized. It is easy to understand. The contribution seems to be good for the task.

Besides, this paper focuses on Human Trajectory Simulation, which is somehow correlated with time series problems essentially. Maybe adding some comparisons with time series methods [1,2] could be better.
[1] DEPTS: Deep Expansion Learning for Periodic Time Series Forecasting.
[2] Dish-TS: a general paradigm for alleviating distribution shift in time series forecasting.

**Questions:**

n/a

**Reviewer Confidence:**

3: The reviewer is confident but not certain that the evaluation is correct

**Scope:**

3: The work is somewhat relevant to the Web and to the track, and is of narrow interest to a sub-community

---

### Official Review · Reviewer_Rab3 · 2023-11-24

**Novelty:** 4
**Technical Quality:** 4

**Review:**

**Pros**

- **Quality** This article conducted detailed experiments on four datasets, comparing 10 baseline models and conducting comprehensive ablation experiments to demonstrate the effectiveness of the proposed method.

- **Clarity** This article is well written and easy to understand.

- **Originality and Significance**
  - This paper attempts to enhance trajectory generation in the target city using knowledge from other cities, which is a unique perspective.
  - This paper draws inspiration from "model-agnostic meta-learning", and this architectural design means that it can be used to enhance the performance of existing models.
  - The proposed method is very helpful for cities with scarce data.

**Cons**
1. **The effectiveness of the baseline model is questionable.** Baseline model MoveSim is a state-of-the-art (SOTA) model published in KDD 20 and has received over 69 citations. The experiments of MoveSim show that its performance on the GeoLife dataset is better than SeqGAN, GAN, DeepMove, etc., but **this article seems to draw the opposite conclusion**, so I have doubts about the experimental results.
2. **The performance improvement brought by the proposed transfer learning pipeline is quite limited**. This paper leverages the mobility data of three source cities to enhances the mobility simulation in the target city, which undoubtedly *requires more resources compared to a single-city baseline*. However, compared to the single city baseline (especially the vanilla transformer), the *performance improvement on the three cities (New York, Singapore, Beijing) are very weak* (at a scale of 10^-4).
3. **Transfer Mobility cross city is not a novel topic.** The experiment part lacks comparison with other transfer learning method for human mobility, such as [1] *What is the human mobility in a new city: Transfer mobility knowledge across cities*, which also published in Web Conference.
4. **The results lack interpretability.** It is unclear what cross city knowledge COLA has acquired to enhance performance (See more detail on **Q2**).
5. **Debatable practical applications (RQ3 & Section 4.4)**. Trajectory simulation is a quite important task intended for application in downstream tasks to avoid privacy issues. This article evaluates it on trajectory prediction tasks, and **the test results show that the generated trajectories are more predictable**. According to the results, *it seems that the simulated trajectories lose the randomness of human real mobility to some degree.* I don’t think this is a good signal. Moreover, the author could consider testing COLA's effectiveness in other task scenarios such as epidemic simulation.
6. This article overlooked some crucial related research.

>    [1] He, T., Bao, J., Li, R., Ruan, S., Li, Y., Song, L., ... & Zheng, Y. (2020, April). What is the human mobility in a new city: Transfer mobility knowledge across cities. In Proceedings of The Web Conference 2020 (pp. 1355-1365).
> [2] Choi, S., Kim, J., & Yeo, H. (2021). TrajGAIL: Generating urban vehicle trajectories using generative adversarial imitation learning. Transportation Research Part C: Emerging Technologies, 128, 103091.

**Questions:**

Q1. In Section 4.4, the performance of three datasets - real, baseline, and COLA - are compared. I am curious about *how the baseline dataset was generated* as it is not explicitly mentioned in the paper.

Q2. Section 4.4 evaluates the performance of simulated trajectory datasets for location prediction tasks and demonstrates that the proposed method produces more predictable trajectories. However, human travel behavior also contains a lot of randomness, so *I'm unsure if the result (i.e., Good Predictability) is a good signal*. What are your thoughts on this?

Q3. The Half-open transformer treats V as private, while Q and K are shareable. I would like to understand why Q and K are shareable. Is this due to some unique characteristics they possess? My assumption is the same as the author: that Q and K model city-universal human mobility patterns. But additional evidence, such as interpretability analysis results, may be necessary to convince me (or other readers) further.

**Reviewer Confidence:**

4: The reviewer is certain that the evaluation is correct and very familiar with the relevant literature

**Scope:**

3: The work is somewhat relevant to the Web and to the track, and is of narrow interest to a sub-community

---

### Official Review · Reviewer_ro6r · 2023-11-25

**Novelty:** 5
**Technical Quality:** 6

**Review:**

**Quality and clarity**

The paper presents an approach, COLA (Cross-city Mobility Transformer), for simulating human mobility across different cities. The quality of the work is good, with a well-structured methodology and clear presentation of results. The authors have successfully demonstrated the application of their model in different cities, showing its adaptability and robustness.

**Originality**

See the questions 1.



**Significance of this work**

This research holds substantial significance, particularly he ability to accurately simulate human mobility patterns can aid for traffic management and epidemic prevention.



**Pros**

1. The methodology is well-documented.
2. Comprehensive experimental evaluation across multiple cities, enhancing the generalizability of the findings.
3. Clear demonstration of the model's ability to outperform existing baselines.

**Cons**

1. While the model performs well, the computational efficiency and scalability were not extensively discussed.

**Questions:**

1. In the introduction, you highlight the motivation stemming from data scarcity (Figure 1) and present domain heterogeneity and long-tail distributions of locations as key challenges. Could you clarify if your work is the first to consider cross-city mobility transfer while explicitly addressing these two challenges? There appears to be a prevalence of studies leveraging cross-city cases. I'm interested in understanding how your approach distinctly contributes to the novelty in this domain. I would check other reviewers' comments and would like to raise my score if the authors claim this well.
2. Your manuscript does not clearly differentiate the unique challenges in your setting compared to other spatio-temporal transfer methods across cities, such as those used in RegionTrans (Line 205) and TrafficTL (Line 210). These methods also seem to account for traffic and location-specific factors. Could you elucidate the specific aspects that set your approach apart from these existing methodologies?
3. It appears that your comparative analysis primarily focuses on a single baseline employing a Transformer model. Are there other baselines leveraging the Transformer architecture?

**Reviewer Confidence:**

2: The reviewer is willing to defend the evaluation, but it is likely that the reviewer did not understand parts of the paper

**Scope:**

3: The work is somewhat relevant to the Web and to the track, and is of narrow interest to a sub-community

---

### Official Review · Reviewer_3Lnc · 2023-11-26

**Novelty:** 4
**Technical Quality:** 4

**Review:**

The paper proposed a human trajectory simulation framework based on transfer learning by leveraging multiple different cities to augment the performance on the target city. A transformer based transfer learning model is proposed with shared meta parameters and city specific private parameters. Experiment results show that the proposed framework can outperform most compared schemes on the task.

**Questions:**

1. I wonder if the framework is used in real world, what selection critirias are used to select cities as sources? From ablation studies, e.g., Fig 4, it seems the results od using All source cities only achieve the best performance in terms of half of the metrics. Does that mean it is possible that using additional data from a specific city may even degrade the performance?
2. Is it possible to simply train the model with the data from all cities (i.e., source & target) without any transfer learning pipeline by heuristic location mapping?
3. Since the traditional transformer model on the single city data can already perform very well in Table 2, is there any transformer based baseline with multi-city data for a fair comparison?

**Reviewer Confidence:**

3: The reviewer is confident but not certain that the evaluation is correct

**Scope:**

3: The work is somewhat relevant to the Web and to the track, and is of narrow interest to a sub-community

---

### Decision · Program_Chairs · 2024-01-22

**Decision:**

Accept

**Comment:**

The paper proposed a human trajectory simulation framework based on transfer learning by leveraging multiple different cities to augment the performance on the target city. Reviewers acknowledge that the paper is well written, and the experiments are comprehensive to show the clear improvement.